# Photoluminescence and Scintillation Properties of Ce^3+^-Doped GdBO_3_ Nanoscintillator Sensors: Effect of Some Synthesis Parameters

**DOI:** 10.3390/mi17010034

**Published:** 2025-12-28

**Authors:** Lakhdar Guerbous, Mourad Seraiche, Ahmed Rafik Touil, Zohra Akhrib, Rachid Mahiou

**Affiliations:** 1Physics Division, Algiers Nuclear Research Center (CRNA), 02, Bd Frantz Fanon, BP 399, Algiers 16000, Algeria; 2Faculty of Sciences, Med Boudiaf University-M’sila, BP 166, M’sila 28000, Algeria; mourad.seraiche@univ-msila.dz; 3Research Center in Industrial Technologies CRTI, P.O. Box 64, Cheraga, Algiers 16014, Algeria; r.touil@crti.dz; 4Department of Physics, Faculty of Science, Ferhat Abbas University of Setif 1, Setif 19000, Algeria; akhrib.zohra@univ-setif.dz; 5Institut de Chimie de Clermont-Ferrand, Université Clermont Auvergne, CNRS, SIGMA Clermont, F-63000 Clermont-Ferrand, France; rachid.mahiou@univ-bpclermont.fr

**Keywords:** GdBO_3_:Ce^3+^, nanoscintillator, sensor

## Abstract

Cerium (Ce^3+^)-doped gadolinium orthoborate (GdBO_3_) phosphor powders were synthesized via an aqueous sol–gel route, with systematic variation in solution pH (2, 5, and 8) and annealing temperature (600–1200 °C, in 100 °C increments) to investigate their influence on structural, optical, and scintillation properties. The materials were comprehensively characterized using thermogravimetric and differential thermal analysis (TG–DTA) to assess thermal behavior, X-ray diffraction (XRD) for crystal structure determination, Fourier-transform infrared spectroscopy (FTIR) for vibrational analysis, and both photoluminescence (PL) and radioluminescence (RL) spectroscopies to evaluate optical and scintillation performance. All samples crystallized in the hexagonal GdBO_3_ vaterite phase (space group P6_3_/mcm). The PL and RL emission spectra were consistent with the Ce^3+^ 5d–4f transitions, and scintillation yields under X-ray excitation were quantified relative to a standard Gadox phosphor. A decrease in photoluminescence quantum yield (PLQY) was observed at annealing temperatures above 800 °C, which is attributed to the incorporation of Ce^3+^ into the host lattice. Scintillation decay profiles were recorded, enabling extraction of timing kinetics parameters. Overall, the results reveal clear correlations between synthesis conditions, structural evolution, and luminescence behavior, providing a rational basis for the optimization of Ce^3+^-doped GdBO_3_ phosphors for scintillation applications.

## 1. Introduction

Inorganic nanomaterial phosphors doped with rare-earth elements have garnered significant attention due to their valuable properties and wide range of potential applications. Nanomaterials typically possess a higher surface area-to-volume ratio compared to bulk materials, which often leads to different properties. Moreover, precise control over the crystallographic structure, size, shape, and dimensionality of nanocrystals is essential, as these factors strongly influence their optical characteristics. Sensors based on scintillator nanomaterials fall into this category and operate by detecting ionizing radiation through the material’s capacity to convert absorbed energy into luminescent light. These sensors are used in diagnostic and therapeutic medical devices, as well as in high-energy detection systems. The rapid progress of technologies in these fields has led to a growing demand for high-performance scintillators. Gadolinium orthoborate (GdBO_3_) is recognized as an excellent host material for rare-earth ions, because of its high transparency in the vacuum ultraviolet (VUV) region and its remarkable resistance to optical damage [1,2,3]. GdBO_3_ crystallizes in the vaterite phase with P63/mmc as the space group; however, it is important to note that there is currently no consensus regarding its crystallographic structure [4]. Additionally, the trivalent rare-earth ion cerium (Ce^3+^) serves as an important optical activator. Possessing a single optically active electron, it has the simplest energy-level structure among all lanthanide ions. Moreover, its strong luminescence arises from spin-allowed interconfigurational 5d → 4f transitions, making Ce^3+^ an excellent activator for scintillator materials used in a wide range of applications. Furthermore, it is widely recognized that the synthesis route plays a crucial role in determining the physicochemical characteristics of the resulting nanomaterials. Indeed, rare-earth doping and co-doping GdBO_3_ micro- and nanophosphors have been synthesized through various methods, including solid-state reaction, hydrolysis, the ultrasonic spray process, sol–gel, and precipitation techniques. Various GdBO_3_-based nanomaterials have been synthesized through doping and co-doping with different trivalent rare-earth ions, including GdBO_3_: Eu^3+^ [5], GdBO_3_/silica:Ce^3+^ nanocomposites [6,7], and core/shell-type GdBO_3_ systems doped with Ce^3+^ and Tb^3+^ [6]. The sol–gel method provides high purity, precise compositional control, versatility in shaping, and low processing temperatures, making it highly beneficial for the synthesis of advanced materials, especially in fields such as nanotechnology, optics, catalysis, and coatings. It is well established that controlling the synthesis conditions, like the pH value and annealing temperature, in soft chemistry methods has a significant impact on the crystallin structure, morphology, and, consequently, the spectroscopic properties of the resulting nanomaterials. Although numerous studies have reported on GdBO_3_ doped with rare-earth ions such as Eu^3+^ and Pr^3+^, to the best of our knowledge, no investigations have addressed the synthesis of Ce^3+^-doped GdBO_3_ via an aqueous sol–gel method.

In this context, the present work represents the first systematic study of GdBO_3_: Ce^3+^ synthesized via the proposed sol–gel route. This study focuses on the preparation of nanometric GdBO_3_:Ce^3+^ powders and provides an initial evaluation of their spectroscopic properties. Furthermore, the radioluminescence performance of the synthesized materials is benchmarked against that of the commercial Gadox:Tb^3+^ phosphor, highlighting their potential relevance for scintillation applications. This work builds on our previous research on rare-earth-doped gadolinium borates (GdBO_3_), as cited in the bibliography. Here, Ce^3+^-doped GdBO_3_ vaterite-type nanophosphors were successfully produced using an acetylacetone-assisted aqueous sol–gel method. A systematic investigation was conducted to examine the effects of solution pH and annealing temperature on the structural, vibrational, photoluminescence, and radioluminescence properties of the materials.

## 2. Materials and Methods

Ce^3+^ (xCe = 0.5 mol%)-doped gadolinium borate GdBO_3_ nanophosphors were prepared by a simple aqueous sol–gel process. First, stoichiometric amounts of high-purity gadolinium (III) nitrate hexahydrate Gd(NO_3_)_3_.6(H_2_O) (99.9%), cerium (III) nitrate hexahydrate Ce(NO_3_)_3_.6(H_2_O) 99.9%, and boric acid H_3_BO_3_ (99.5%) used as source of boron, were dissolved in high-quality de-ionized water (resistivity ≳ 18.2 MΩ·cm at 25 °C). In order to compensate for the volatilization of B_2_O_3_ at a high temperature during the heating treatment, an excess of 15 mol% of H_3_BO_3_ was added to the raw materials. The mixed solution was stirred at room temperature for 1 h, resulting in a transparent solution. Then, acetylacetone (CH_3_COCH_2_COCH_3_) (99%) was added as a chelating agent at a 1:1 molar ratio relative to the product. Acetylacetone (AC) is introduced to stabilize Gd^3+^ and Ce^3+^ cations through coordination via its oxygen atoms. This chelation moderates the hydrolysis and condensation kinetics, leading to the formation of a more homogeneous sol and gel network and promoting the uniform incorporation of the Ce^3+^ dopant. In addition, acetylacetone contributes to the control of crystallite growth and size. Consequently, by improving the dopant distribution and microstructural homogeneity, acetylacetone indirectly enhances the luminescence efficiency and reproducibility of Ce^3+^-doped materials while reducing defect-related non-radiative quenching centers. To stabilize the solution, stirring was carried out for 30 min. After that, its pH value was adjusted to 2, 5, and 8 by adding ammonium hydroxide (NH_4_OH) (99.9%). The solution was then stirred an extra hour, until a homogeneous and transparent solution was obtained. The resulting solution was dried at 80 °C and stirred until transforming to a xerogel. Finally, the xerogel for each pH variant was annealed at 900 °C for 4 h. In addition, different annealing temperatures (ranging from 600 °C to 1200 °C) were applied to the xerogel corresponding to the solution with pH = 8. The procedure used in this study for synthesis is shown in a flow diagram in Figure 1.

Phase identification was performed, and related structural properties of nanopowders were investigated by the X-ray diffraction (XRD) technique using a PAN analytical X′Pert (Philips) PRO (PANalytical B.V., Almelo, The Netherlands) apparatus with CuKα radiation (λ = 1.54059 Å) operated at 45 kV and 40 mA. Symmetric (θ-θ) scans were performed from 10° to 90° (2θ) with a step width of 0.02°. All the data were processed by X′Pert High Score plus Software (version 3.0) with commercial databases (FWHM deduction and peak identification). The lattice parameters, average crystallite size, and macrostrain were estimated from the X-ray diffraction patterns. The crystallite size and the lattice strain were evaluated using the Williamson–Hall (W–H) relationship [8].(1)$$\beta \times \frac{\cos\theta }{\lambda }=\frac{1}{D}+\eta \sin\theta /\lambda$$
where β represents the full width at half maximum, λ is the X-ray wavelength, θ denotes the diffraction angle, D is the effective crystallite size, and η indicates the effective strain. In the Williamson–Hall method, it is assumed that the strain is isotropic and the crystallite size is uniform. With plot of $\beta \times \frac{\cos\theta }{\lambda }$ against sinθ/λ, and a linear fit, the intercept provides the crystallite size, while the slope indicates the microstrain. The standard error in the intercept of the W–H plot was used to estimate the uncertainty in crystallite size and microstrain.

Thermogravimetric analysis (TG) and differential thermal gravimetry (DTA) were performed by SETARAM Setsys Ev 1750 (TGA-DTA 1600) instrumentation (SETARAM Instrumentation, Caluire, France). The infrared spectra were recorded in the range of 400–4000 cm^−1^ with a Nicolet-IR 380 Fourier-transform infrared (FTIR) spectrometer (Thermo Fisher Scientific, Madison, WI, USA). The room-temperature steady photoluminescence spectra were generated using a Perkin-Elmer (LS-50B) (Perkin-Elmer Inc., MS, Shelton, CT USA) luminescence spectrometer with Xe lamp excitation. The photoluminescence quantum yield (QY) values were measured using a Hamamatsu PL-QY C9920-02G system (Hamamatsu Photonics, Shizuoka, Japan). This instrument is equipped with a 150 W monochromatic xenon lamp, an integrating sphere coated with Spectralon (diameter: 3.3 inches), and a high-sensitivity CCD spectrometer, allowing detection of the full luminescence spectrum. The radioluminescence spectra were recorded at 300 K under X-ray irradiation. The excitation source was an X-ray tube (Philips 2274) operated at 30~kV, 20~mA. A monochromator working in the range of 430–770 nm and an ANDOR CCD camera were used. Scintillation decay measurements were performed at 300 K under X-ray excitation using a picosecond pulsed laser (C10196; Hamamatsu Inc., Iwata, Japan), with 100 kHz and 0.2 mA, and YG11 interference filter. The TCSPC module (PicoHarp 300) was adopted to analyze the temporal histogram for the counts.

## 3. Results

### 3.1. TDA and TG Analysis

Thermal analysis was used to study the evolution of GdBO_3_ crystal growth, from its initial amorphous xerogel form to its crystallized state. The corresponding thermogram of GdBO_3_: Ce^3+^ for pH = 8 taken as example is shown in Figure 2. To better identify the temperatures associated with various mass loss events during the thermal treatment of GdBO_3_ (ranging from 0 °C to 1100 °C), the first derivative of the thermogram is also presented.

The thermogravimetric (TG) curve, highlighted in blue in Figure 2, shows three distinct mass loss regions. The first two regions occur between 80 °C and 150 °C, and between 230 °C and 300 °C, and are assigned to the elimination of adsorbed species such as water or alcohol molecules. A significant second mass loss occurs between 210 °C and 300 °C, corresponding to the condensation step of GdBO_3_. This phenomenon, with a strong exothermic peak at around 282 °C on the DTA curve (black curve), is probably associated with the evaporation and decomposition of water and residual organic matter. Between 210 °C and 700 °C, the formation of intermediate inorganic compounds takes place. A third mass loss is observed between 300 °C and 490 °C, linked to the initial crystallization phase of GdBO_3_. This final mass loss corresponds to the growth of the crystalline GdBO_3_ lattice and the removal of remaining hydroxyl groups. No further mass loss is observed above 700 °C. The total mass loss recorded up to 700 °C is approximately 7%. Based on the thermal behavior observed, optimal crystallization temperatures were determined to be above 1000 °C. In particular, the formation of GdBO_3_ with a hexagonal structure begins above 700 °C, as indicated by the nearly linear and stable region of the DTA curve between 700 °C and 1100 °C.

### 3.2. XRD Analysis


**pH effect**


The XRD patterns of GdBO_3_:0.5% mol Ce^3+^ powders synthesized with different pHs and annealed at 900 °C/4 h are shown in Figure 3. All diffraction peaks for each pH value are attributed to the pure hexagonal structure for the GdBO_3_ phase and match well with the ICDD No. 74-1929 card with space group P63/mcm. No other phase was present. The lattice parameters calculated from the diffraction patterns are listed in Table 1.

It can be observed that the average crystallite size decreases, while the microstrain increases with an increasing solution pH. In fact, the sol–gel process involves two main reaction mechanisms: hydrolysis and condensation. At a low pH (acidic conditions), the reaction is primarily driven by hydronium ions (H_3_O^+^) present in the solution. The rapid reaction between water and protons (H_2_O + H^+^ → H_3_O^+^) leads to a reduced availability of water in the final product. In contrast, at a high pH (basic conditions), hydroxide ions (OH^−^) govern the reaction mechanism. Typically, under acidic conditions, condensation of hydroxide species tends to form linear polymer chains, resulting in smaller crystallite sizes [9].

Conversely, under basic conditions, the formation of larger, branched polymer networks generally leads to larger crystallite sizes. However, in our case, the opposite behavior is observed: increasing the pH results in smaller crystallite sizes. This suggests that, rather than OH^−^ ions, H_3_O^+^ ions may still play a dominant role in the reaction mechanism even at higher pH levels. More broadly, the observed decrease in crystallite size with an increasing pH is likely associated with changes in precursor complexation, hydrolysis–condensation equilibria, and nucleation density inherent to rare-earth sol–gel systems, rather than with the dominance of a single ionic species.

Furthermore, the absence of water-related absorption bands in the FTIR spectra (Section 3.3) confirms that water molecules are largely eliminated during the annealing process, reinforcing the proposed mechanism. The structural ratio c/a increases with the increment of pH value, due to an increase in the c parameter, while a remains constant. As a result, the crystal structure expands along the c-axis with an increasing pH.


**Annealing temperature effect**


Figure 4 shows the X-ray diffraction patterns of GdBO_3_ doped with 0.5% Ce^3+^ (pH = 8), calcined at various temperatures (600, 700, 800, 900, 1000, 1100, 1200 °C) for 4 h. The X-ray diffractograms exhibit similar patterns, all corresponding to the vaterite structure, with P63/mmc as the space group, regardless of the calcination temperature. Notably, no secondary or parasitic phases were detected. However, an increase in the intensity of the diffraction peaks is observed with a rising calcination temperature, indicating improved crystallinity.

According to Table 2, both the crystallite size and strain exhibit irregular variations with the calcination temperature. As the calcination temperature was raised from 600 °C to 700 °C, the crystallite size increased from 47 nm to 92 nm. However, a subsequent decrease was observed between 700 °C and 800 °C. Between 900 °C and 1200 °C, an increment in crystallite size was observed, accompanied by an increase in lattice strain. Unexpectedly, the sample annealed at 900 °C exhibits a smaller crystallite size than those annealed at other temperatures, with the exception of the 600 °C sample, despite all samples being prepared under identical conditions and annealed in the same furnace. It is well established that the growth of nanomaterials during heat treatment processes such as annealing, calcination, and sintering is governed by several interrelated thermodynamic and kinetic factors. These factors influence particle coalescence, grain growth, and phase transformations. Generally, higher temperatures favor larger crystallite sizes while reducing the density of structural defects. This unexpected behavior warrants further investigation.

### 3.3. FTIR Analysis


**pH effect**


The FTIR spectra of GdBO_3_: 0.5 mol% Ce^3+^ samples synthesized at pH values of 2, 5, and 8 are shown in Figure 5. As can be observed, the FTIR spectra of GdBO_3_ doped with 0.5 mol% Ce^3+^ at pH values of 2, 5, and 8 are similar. The obtained phase is identified as pure vaterite, which is characterized by the presence of B_3_O_9_^9−^ groups [10,11]. All detected bands correspond to characteristic vibrational modes: the bending vibrations of O-B-O bonds (δ O-B-O) in the range of 600–900 cm^−1^, the stretching vibrations of Gd-O bonds (ν Gd-O) between 400 and 600 cm^−1^, and the B-O stretching vibrations (ν B-O) in the 900–1100 cm^−1^ region [12,13]. Overall, no significant changes were observed with an increasing pH value.


**Annealing temperature effect**


The FTIR spectra of GdBO_3_: 0.5 mol% Ce^3+^ samples (pH = 8) in xerogel as well as treated at different temperatures for 4 h are shown in Figure 6. One can observe that the FTIR spectra of all calcined samples are similar. Annealing treatment at between 600 °C and 800 °C leads to the elimination of organic residues from the xerogel, which are completely removed at 900 °C. Only the bands corresponding to the vibration frequencies of Gd-O bonds near 570 cm^−1^ and B-O bonds in the (B_3_O_9_^9−^) groups are observed within the 500 to 1250 cm^−1^ range.

### 3.4. Steady Excitation and Emission Photoluminescence Study


**Excitation spectrum**


All GdBO_3_: 0.5 mol% Ce^3+^ nanopowder samples recorded at room temperature exhibit similar excitation spectra, as shown in Figure 7. The room-temperature excitation spectrum, measured for λ_em_ = 420 nm, exhibits three main broad bands, with maxima situated at 357 nm, 340 nm, and 235 nm. These bands were assigned to the 4f^1^ (^2^F_5/2_) ground state, to its field-split 5d levels in GdBO_3_ [13]. It is important to note that in the excitation spectra of GdBO_3_:Ce^3+^, no energy levels corresponding to Gd^3+^ are observed, indicating the absence of energy transfer between Gd^3+^ and Ce^3+^ ions.


**pH effect**


Figure 8 shows the emission spectra of GdBO_3_: 0.5 mol% Ce^3+^ nanopowder samples prepared at pH values of 2, 5, and 8 and sintered at 900 °C for 4 h. The emission spectra recorded under 340 nm excitation exhibit two well-resolved, intense broad sub-bands, separated by approximately 1800 cm^−1^. These sub-bands are attributed to the emission transitions from the lowest crystal field-split component of the 5d level to the ground-state doublet of Ce^3+^ (^2^F_5/2_ and ^2^F_7/2_) [13,14]. Furthermore, one can note that the sample prepared at a basic pH (pH = 8) exhibits the highest luminescence intensity.


**Annealing temperature effect**


Since the sample prepared at pH = 8 exhibited the most intense luminescence, it was selected for the study of the effect of the annealing temperature. Figure 9 presents the emission spectra of GdBO_3_: 0.5 mol% Ce^3+^ (pH = 8) samples calcined at various temperatures between 600 and 1200 °C. For each annealing temperature, the emission spectra recorded under 340 nm excitation (corresponding to the 5d absorption of Ce^3+^) display the emission transition bands from the lowest crystal-field-split component of the 5d level to the ground-state doublet of Ce^3+^ (^2^F_5/2_ and ^2^F_7/2_) in the GdBO_3_ host material. Notably, under 340 nm excitation, the Ce^3+^ ion exhibits the maximum emission intensity when calcined at 700 °C for 4 h. However, the emission intensity reaches its maximum for the sample annealed at 700 °C, after which it decreases and then fluctuates at higher temperatures. This variation in emission intensity is likely influenced by several factors, including cross-relaxation between adjacent Ce^3+^ ions due to a further decrease in unit cell parameters, changes in the average nanoparticle size (either increase or decrease), and the formation of additional defects, which may enhance the probability of non-radiative transitions [15,16]. Furthermore, the luminescence intensity can be affected by the competition between Ce^4+^ and Ce^3+^ ions, depending on the annealing temperature [17]. Additionally, a low-intensity band is observed in the 360 nm range, corresponding to the radiative recombination of self-trapped excitons (STEs) [18]. This STE band is more pronounced in the sample annealed at 1000 °C.

The photoluminescence (PL) quantum yield (*QY)* is a key parameter for evaluating the performance of rare-earth-ion-doped luminescent materials. It is defined as the ratio of the number of emitted photons to the number of absorbed photons as follows:(2)$$QY=\;\frac{number\;of\;photons\;emitted}{number\;of\;photons\;absorbed}$$

The variation in *QY* as a function of annealing temperature is also shown in Figure 10. It can be observed that the quantum yields obtained in our study are significantly lower than those reported in the literature. For instance, LuBO_3_:Ce^3+^, Tb^3+^ has been reported to exhibit a quantum yield of approximately 50%, while LuBO_3_:Ce^3+^, Tb^3+^, Eu^3+^ shows a yield close to 30% [19]. At temperatures above 800 °C, the PLQY decrement could be attributed to the integration or bonding of Ce^3+^ to the network.

### 3.5. Steady Radioluminescence Study

The room-temperature X-ray-Excited Luminescence (XEL) spectra of Gd_0.995_Ce_0.005_BO_3_ nanopowders calcined at 900 °C/4 h for different pH values of the solution and annealed at different temperatures for pH = 8 are presented in Figure 11 and Figure 12. Also, the Gadox-Tb (Gd_2_O_2_S:Tb) XEL emission spectrum was recorded, in the same experimental conditions, in order to estimate the scintillation yields. The relative Light Yields (LYs) of all samples were obtained by comparison with that of Gadox-Tb^3+^, with 78,000 photons/Mev [20].


**pH effect**


Figure 11 shows the room-temperature X-ray-Excited Luminescence (XEL) of Gd_0.995_Ce_0.005_BO_3_ nanopowders calcined at 900 °C/4 h for different pH values of the solution. From the figure, one can note that increasing the pH value of the solution increases the XEL intensity, which follows the same behavior as that of the PL intensity (Figure 8). The determined LY values are presented in Table 3. It can be observed that LY increases with an increasing pH value and is closely correlated with the PL emission behavior shown in Figure 8.


**Annealing temperature effect**


Figure 12 exhibits the room-temperature X-ray-Excited Luminescence (XEL) of Gd_0.995_Ce_0.005_BO_3_ nanopowders prepared at pH = 8 and calcined at different temperatures. One can see that as the temperature increases from 600 °C to 800 °C, the XEL intensity increases, and after that, roughly stabilizes. One can note that the evolution of XEL intensity against temperature is different from the PL intensity (Figure 9). In PL spectra, the sample annealed at 700 °C presents the highest emission intensity, whereas, in XEL, samples annealed at temperatures up to 800 °C present the highest intensities. This difference can be related to the greater complexity of the scintillation phenomenon. Knitel et al. [21] report that for Ce^3+^-doped borates such as GdBO_3_, YBO_3_, Li_6_Y(BO_3_)_3_, LaB_3_O_3_, GdB_3_O_6_, LaMgB_5_O_10_, YMgB_5_O_10_, CaLaB_7_O_13_, and Ca_4_YO(BO_3_)_3_, the scintillation light yield ranges from approximately 6000 to 10,000 photons/MeV. An exception is observed in scintillator materials without gadolinium, which plays an important role in the efficient transfer of excitation energy via Gd^3+^ to Ce^3+^.

The observed variations in light yield (LY), together with the differences in photoluminescence (PL) and X-ray excited luminescence (XEL) intensities, where PL exhibits a maximum at 700 °C and XEL at 800 °C, suggest that the relationship between PL and XEL behavior is not straightforward. These discrepancies may originate from distinct excitation mechanisms, differences in penetration depth, or non-radiative processes that are not fully reflected in PL measurements alone.

### 3.6. Time-Resolved Radioluminescence

As it is interesting to investigate the effect of rise and decay times on the timing resolution for nanoscintillator materials [22], the scintillation time profiles of all samples presented in Figure 13 and Figure 14 were fitted to some of the experimental functions to determine the rise and decay times, as presented in Figure 11 (inset: we also present an example of a fitted decay curve). The extracted values of the rise time, decay time, and average decay time are summarized in Table 4 and Table 5. These data correspond to samples prepared under different pH values and annealing temperatures. It can be observed that the sample prepared at pH = 5 exhibits the fastest rise time, and the average decay times decrease with an increasing pH value of the solution. For different pH values, the faster components exhibit higher intensities than the slower ones. Furthermore, from Table 5, one can note that the rise time increase from 0.25 ns for 600 °C and reaches its maximum for the sample annealed at 900 °C with 0.90 ns; after that, it decreases to 0.76 for 1200 °C. In fact, a shorter rise time and faster decay indicate more efficient energy transfer from thermalized pairs to Ce^3+^ ions, resulting in a higher transfer efficiency. In contrast, slow or multi-component rise and decay times indicate competition with other processes, such as energy transfer to different centers, or traps, which can decrease the overall efficiency of energy transfer to the intended Ce^3+^ state. Additionally, when some Ce^3+^ ions are oxidized to Ce^4+^, the quantity of Ce^3+^ ions available for energy reception diminishes. This reduces the transfer efficiency, as energy that would normally excite Ce^3+^ ions can instead be lost through non-radiative processes associated with Ce^4+^ ions, resulting in a longer rise time. For comparison, Zhang et al. [4] reported a Ce^3+^ decay time of 24.9 ns for GdBO_3_:Ce^3+^.

This light yield (LY) depends on the number of photons *N_Ph_* produced in the scintillator material under X-ray excitation [23]. In addition, the light yield (*LY*) is influenced by the light collection efficiency (*η_coll_*), which accounts for all optical losses encountered throughout the propagation path to the photodetector, such as reflection, absorption, and scattering within the nanoscintillator powder material and through the competition between these losses and light extraction at the powder surface [23]. This parameter depends on the optical properties of the nanoscintillator, especially the refractive index. The number of photons produced (*Nₚₕ*) depends on the overall efficiency of converting incident X-ray energy into UV or visible light and is given by the following equation [23]:(3)$$N_{ph}=\;\frac{E_{i}}{\beta E_{g}}\times S\times Q$$
where *E_i_* is the energy of incident X-ray radiation, *E_g_* is the bandgap energy, *S* is the transfer efficiency from thermalized pairs to the excited states of luminescent centers (Ce^3+^), and *Q* is the efficiency of luminescence (Figure 10). *β* is a parameter varying from 2 to 3 for a wide range of scintillators [23]. Thus, the nanoscintillator light yield is defined by the following equation:(4)$$LY=\;\eta_{coll}\frac{S\times Q}{\beta E_{g}}$$

If we consider $\eta_{coll}$ to be the same for all samples, we can determine the transfer efficiency normalized to the light collection efficiency $\eta_{coll}$ (S/$\eta_{coll}$) ratio as(5)$$S/\eta_{coll}=LY\times \frac{\beta E_{g}}{Q}$$

For GdBO_3_, the band gap is around 7.1 eV [3,13,24]. Figure 15a shows the variation in the *S/η_coll_* ratio, as well as the rise and average decay times, with respect to the annealing temperature. As can be observed, all of these parameters exhibit temperature dependence. For temperatures in the 600 to 800 °C range, all these quantities increased, and from 800 to 1200 °C, the rise and decay times roughly evolved in the opposite direction of the *S/η_coll_* ratio. In fact, when the transfer efficiency is high, excitons or charge carriers reach the luminescent centers more rapidly, leading to a shorter rise time. Overall, these results indicate that the transfer efficiency improves as the annealing temperature increases. The schematic shown in Figure 15b illustrates the excitation energy transfer underlying the scintillation and photoluminescence mechanisms in GdBO_3_: Ce^3+^.

The coincidence time resolution (CTR) is closely linked to the radioluminescence kinetics, particularly their rise (*τ_r_*) and decay (*τ_d_*) times and the light yield. It can be approximated by the following relation [25,26]:(6)$$CTR=2.18\times \;\sqrt{\frac{\tau_{r}\tau_{d}}{LY}}$$

Table 4 and Table 5 summarize the determined CTR values as a function of the pH and annealing temperature, respectively. A time-of-coincidence resolution (CTR) of less than 100 ps in time-of-flight positron emission tomography is expected to significantly enhance image quality [27]. It can be observed that the CTR values of all samples exceed 100 ps, with the exception of the sample annealed at 1100 °C. Moreover, a general improvement in CTR can be observed with an increasing annealing temperature, leading to an enhanced temporal resolution.

## 4. Conclusions

Phase-pure vaterite GdBO_3_ phosphors doped with Ce^3+^ ions were successfully synthesized for the first time using an acetylacetone-assisted aqueous sol–gel method, providing a simple and cost-effective route. Systematic studies were conducted across various pH values and annealing temperatures, revealing that both factors significantly influence the structural and spectroscopic properties. For all pH values and annealing temperatures, the samples exhibit a pure hexagonal vaterite structure with the P63/mmc space group. The crystallite sizes were found to vary between 47 nm and 93 nm, depending on the solution pH and annealing temperature. These findings were corroborated by the FTIR analysis, which showed the presence of only the B_3_O_9_^9−^ group. Also, the FTIR results indicate that variations in solution pH and annealing temperature during the sol–gel process do not lead to significant changes in the short-range connectivity of the borate network. Broad emission bands were observed under both direct excitation (UV) of the Ce^3+^ ions and X-ray excitation. These bands are attributed to interconfigurational electric dipole transitions, which are allowed due to the 4f–5d parity of Ce^3+^ in GdBO_3_ vaterite. It was observed that the photoluminescence (PL) emission intensity increases with pH. Additionally, the PL quantum yield (PL-QY) rises between 600 and 800 °C, then decreases, mirroring the variation in PL emission with annealing temperature. The scintillation parameters—namely the light yield, decay and rise times, S/η_coll_ ratio, and coincidence time resolution (CTR)—were determined and discussed. The observed discrepancies in radioluminescence (RL) yield, as well as the differing trends between photoluminescence (PL) and X-ray-excited luminescence, may result from distinct excitation mechanisms, variations in penetration depth, or non-radiative processes that are not fully captured by PL measurements alone. The coincidence time resolution (CTR) values of all samples exceed 100 ps, except for the sample annealed at 1100 °C, and a general improvement in CTR is observed at higher annealing temperatures, resulting in an enhanced temporal resolution.

Future research will be directed toward a direct investigation of the Ce^3+^/Ce^4+^ ratio, defect states, and cross-relaxation mechanisms using complementary techniques such as X-ray photoelectron spectroscopy (XPS), electron paramagnetic resonance (EPR), and thermoluminescence. These studies are expected to provide quantitative mechanistic insights, thereby reinforcing the causal relationships between structural parameters, defect chemistry, and luminescence behavior.

## Figures and Tables

**Figure 1 micromachines-17-00034-f001:**
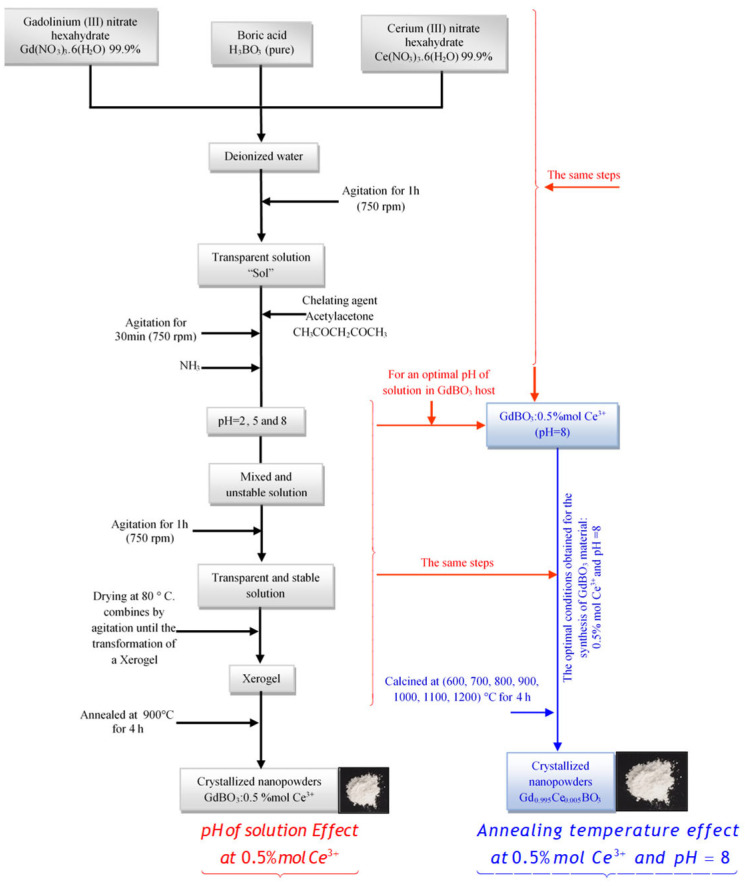
Sequence of steps for synthesizing GdBO_3_: 0.5 mol% Ce^3+^ using the aqueous sol–gel method.

**Figure 2 micromachines-17-00034-f002:**
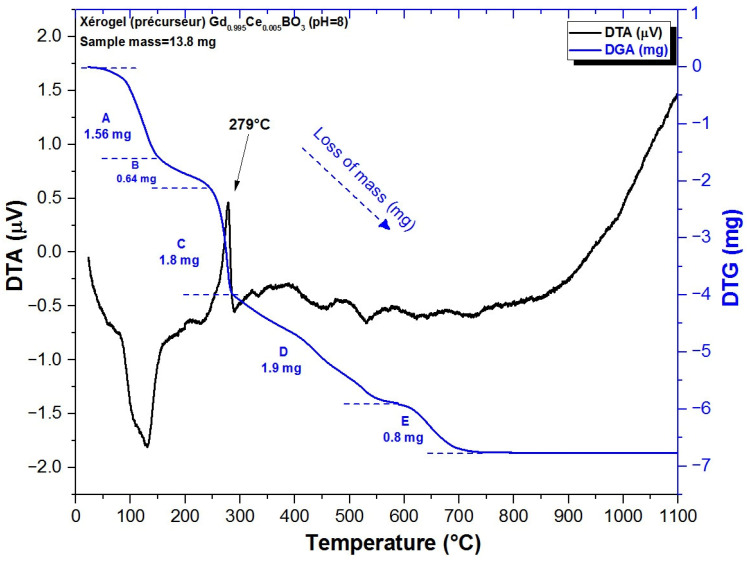
DTG and DTA curves of the Gd_0.995_Ce_0.005_BO_3_ (pH = 8) precursor, obtained by aqueous sol–gel process. A–E indicate indicative of mass-loss regions.

**Figure 3 micromachines-17-00034-f003:**
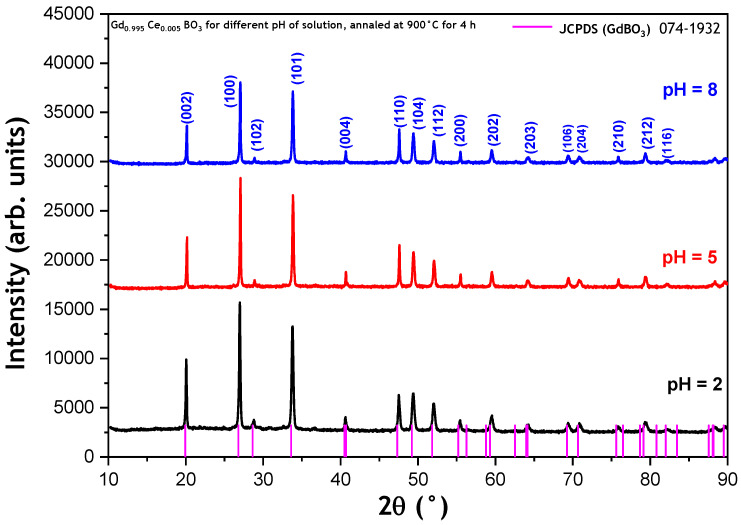
XRD patterns of GdBO_3_: 0.5 mol% Ce^3+^ annealed at 900 °C/4 h for different pH values of solution.

**Figure 4 micromachines-17-00034-f004:**
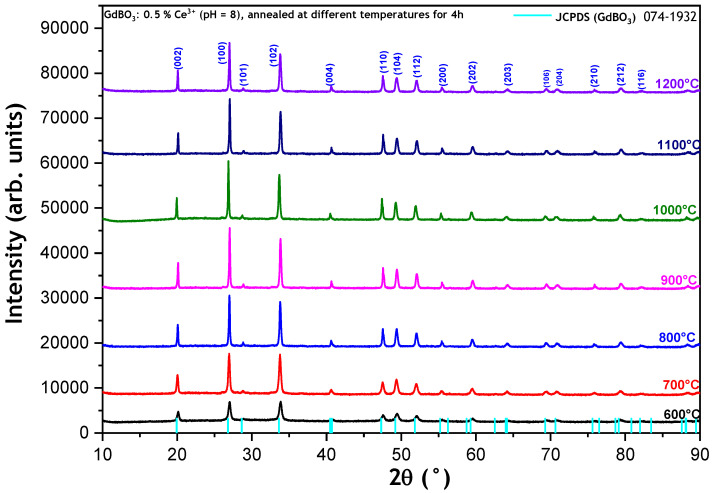
X-ray diffraction patterns of GdBO_3_ powders doped with 0.5% Ce^3+^ (pH = 8), calcined at various temperatures (600–1200 °C) for 4 h.

**Figure 5 micromachines-17-00034-f005:**
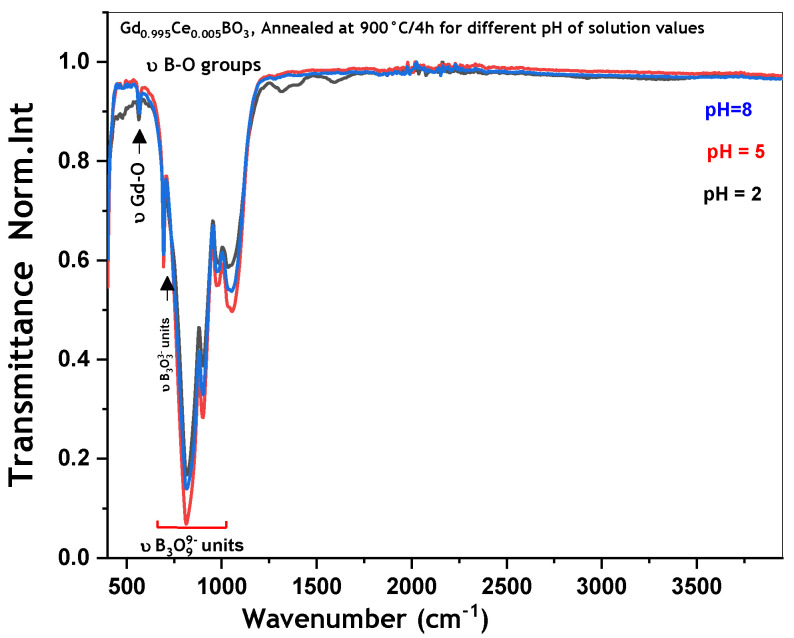
FTIR spectra of Gd_0.995_Ce_0.005_BO_3_ annealed at 900 °C/4 h for different pH values of solution.

**Figure 6 micromachines-17-00034-f006:**
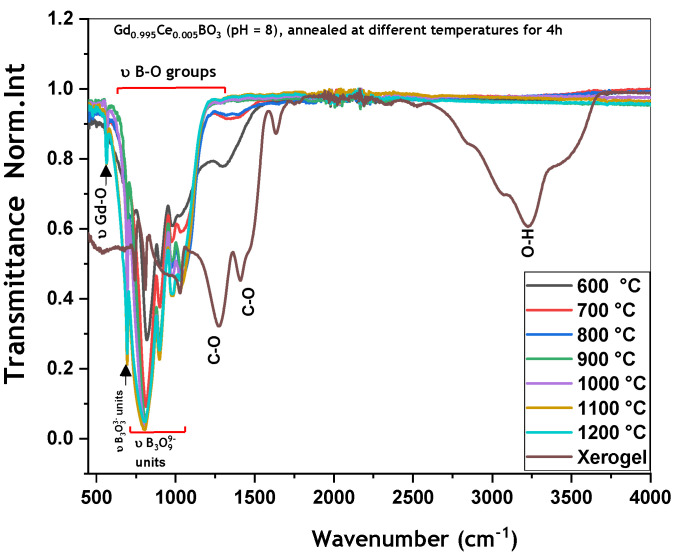
FTIR spectra of Gd_0.995_Ce_0.005_BO_3_ (pH = 8) annealed at different temperatures.

**Figure 7 micromachines-17-00034-f007:**
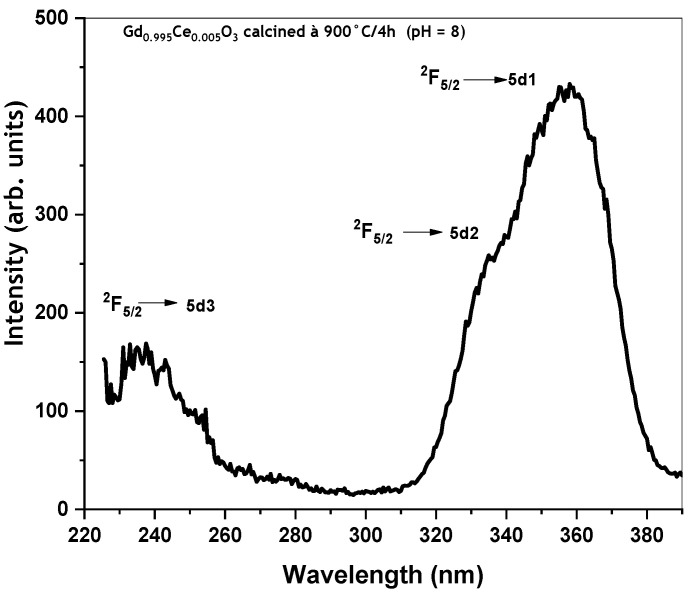
Room-temperature excitation spectrum of GdBO_3_: 0.5 mol% Ce^3+^ (pH = 8) annealed at 900 °C/4 h for λ_em_ = 420 nm.

**Figure 8 micromachines-17-00034-f008:**
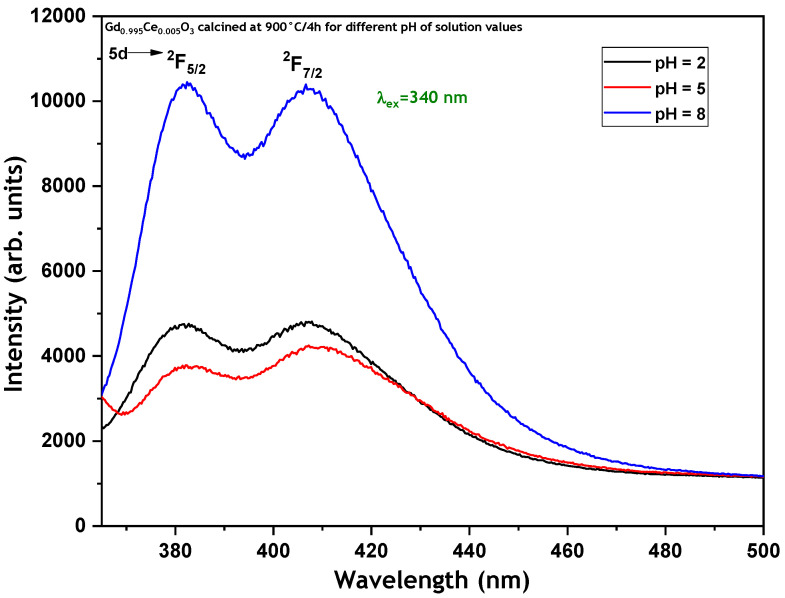
Emission spectra of GdBO_3_: 0.5 mol% Ce^3+^ samples prepared at pH = 2, 5, and 8, and annealed at 900 °C for 4 h, under λ_ex_ = 340 nm.

**Figure 9 micromachines-17-00034-f009:**
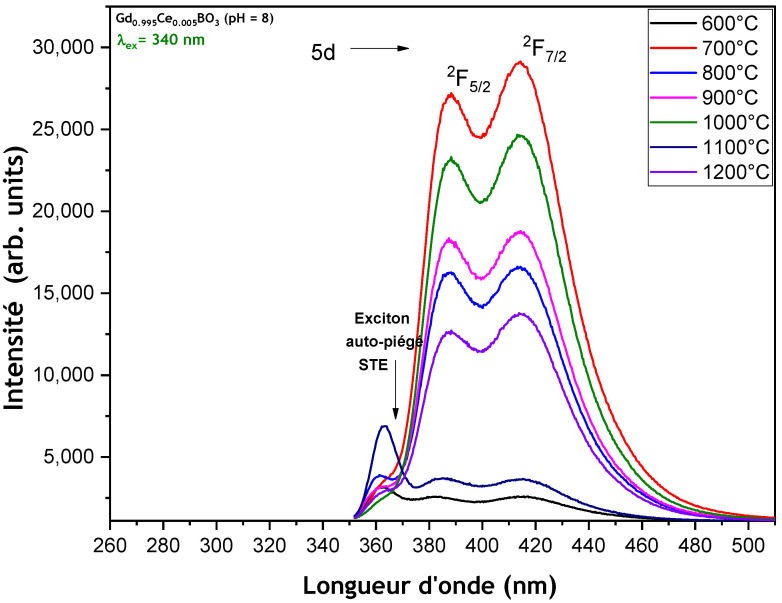
Emission spectra of GdBO_3_: 0.5 mol% Ce^3+^ (pH = 8) sample powders calcined at different temperatures for 4 h under λ_ex_ = 340 nm.

**Figure 10 micromachines-17-00034-f010:**
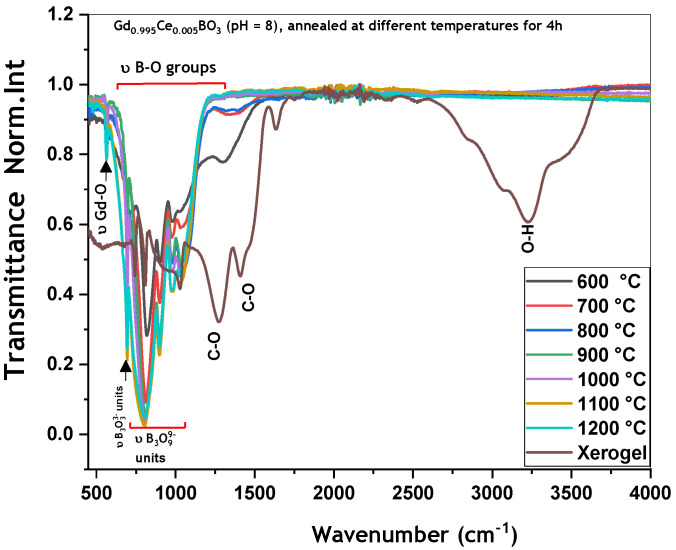
Quantum yield GdBO_3_: 0.5% Ce^3+^ (pH = 8) powders according to the different annealing temperatures.

**Figure 11 micromachines-17-00034-f011:**
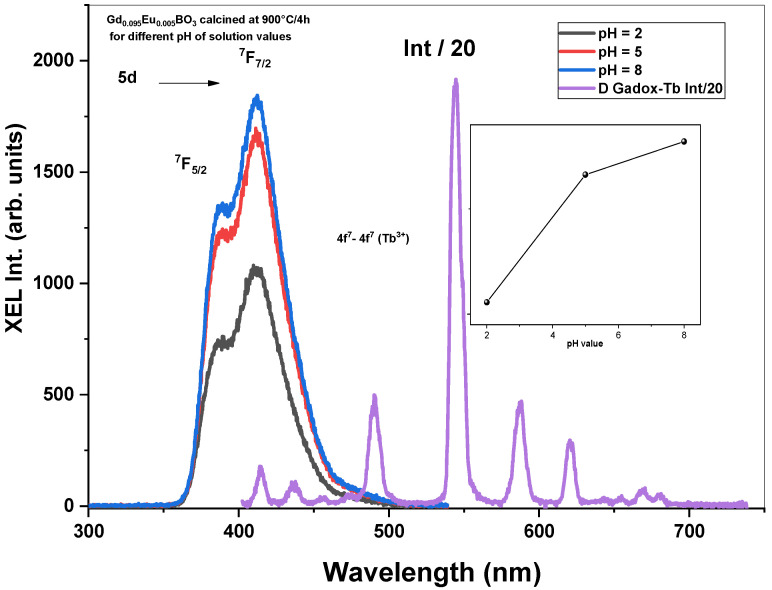
Radioluminescence spectra of Gd_0.995_Ce_0.005_BO_3_ nanopowders calcined at 900 °C/4 h for different pH values of solution.

**Figure 12 micromachines-17-00034-f012:**
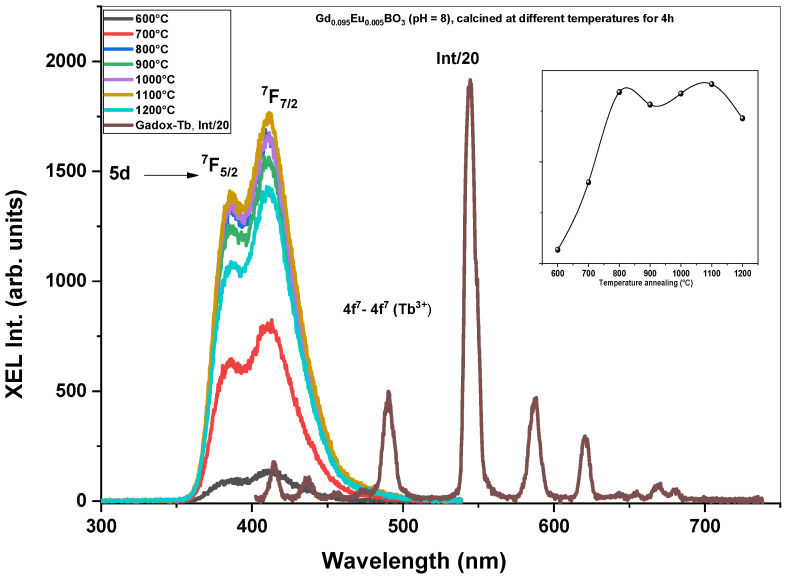
Radioluminescence spectra of Gd_0.995_Ce_0.005_BO_3_ nanopowders prepared at pH = 8 calcined at different temperatures.

**Figure 13 micromachines-17-00034-f013:**
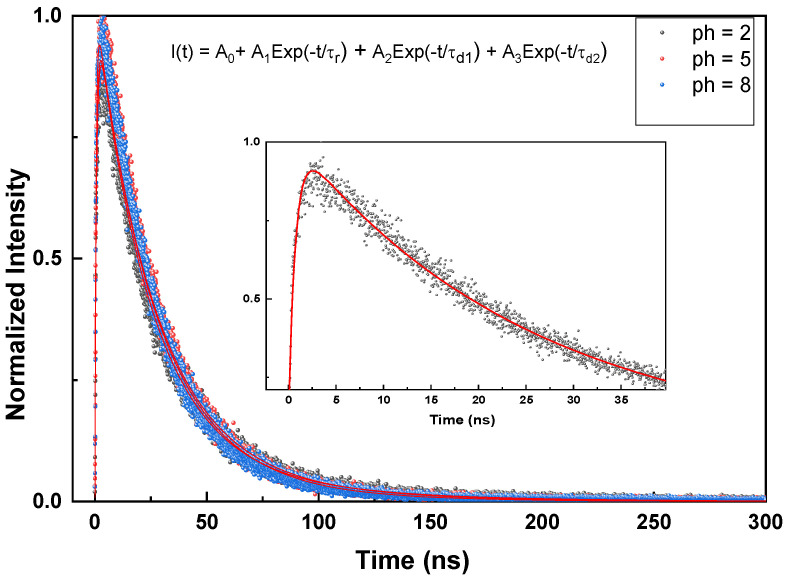
Scintillation decay time profile of Gd_0.995_Ce_0.005_BO_3_ prepared at different pH values of solution. The corresponding exponential lines are likewise presented.

**Figure 14 micromachines-17-00034-f014:**
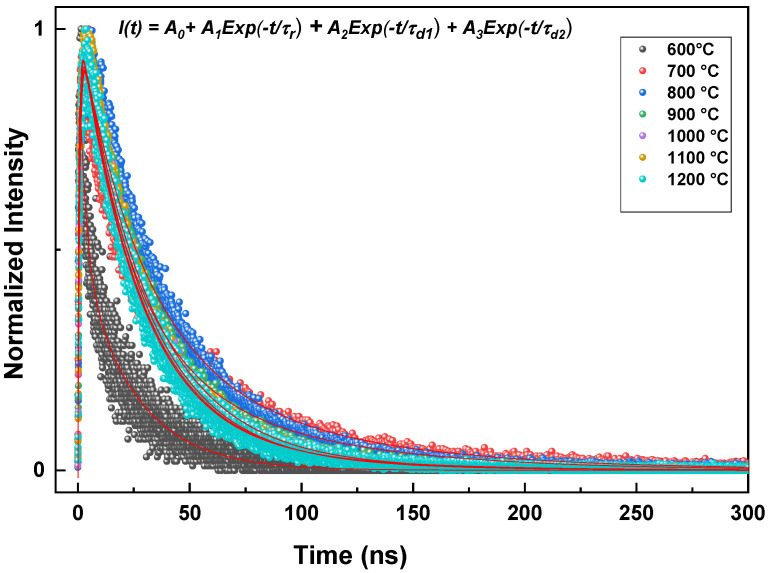
Scintillation decay time profiles for annealing temperature. The corresponding exponential lines are likewise presented.

**Figure 15 micromachines-17-00034-f015:**
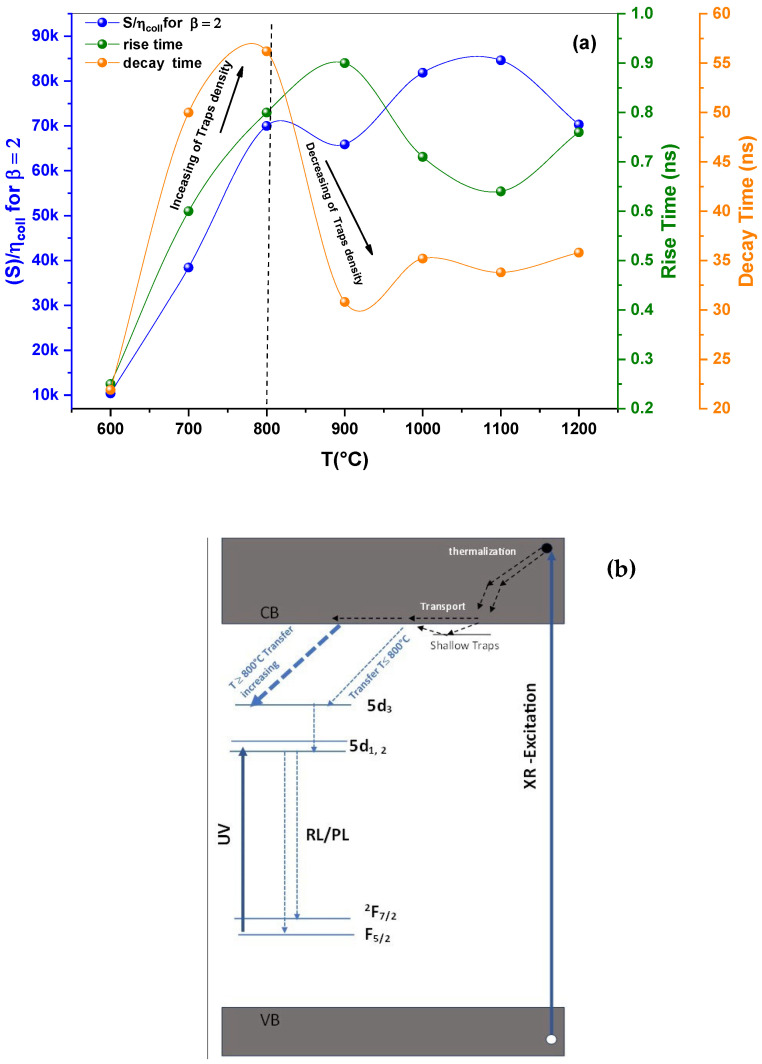
(**a**) Variation in *S/η_coll_* ratio, and the rise and the decay times, as a function of annealing temperature; (**b**) energy level diagram showing excitation energy transfer underlying the scintillation and photoluminescence mechanisms in GdBO_3_: Ce^3+^.

**Table 1 micromachines-17-00034-t001:** Structural parameters of GdBO_3_:0.5%Ce^3+^ powders (pH = 2, 5, and 8) calcined at 900 °C/4 h. The uncertainty is reported in parentheses, consistent with the standard notation for measurement uncertainty.

GdBO_3_:0.5%Ce^3+^ Annealed at 900 °C/4 h	Space Group	Cell Parameters a and b (Å) V (Å^3^)	Crystallite Size (nm)	Strain ×10^−4^
pH = 2	*P63/mmc*	a = b = 3.830 (1)	74.0 (7)	0.447 (4)
		c = 8.892 (4)		
		V = 112.95 c/a = 2.3233		
pH = 5	*P63/mmc*	a = b = 3.825 (2) V = 112.71 c = 8.894 (7) c/a = 2.3238	50.7 (6)	−0.019 (6)
pH = 8	*P63/mmc*	a = b = 3.826 (2) c = 8.899 (7) V = 112.80 c/a = 2.3239	55.5 (6)	−0.012 (9)

**Table 2 micromachines-17-00034-t002:** XRD patterns of GdBO_3_:0.5%Ce^3+^ (pH = 8) calcined at different temperatures for 4 h. The uncertainty is reported in parentheses, consistent with the standard notation for measurement uncertainty.

GdBO_3_:0.5 mol% Ce^3+^ (pH = 8) Annealed at T (°C)	Space Group	Cell Parameters a and b (Å) V (Å^3^)	Crystallite Size (nm)	Strain ×10^−4^
600	*P63/mmc*	a = b = 3.8369 c = 8.9075 V = 113.566 c/a = 2.3221	47.1 (7)	−8.3322 (9)
700	*P63/mmc*	a = b = 3.8348 c = 8.9089 V = 113.452 c/a = 2.3229	92.5 (9)	0.0020 (0)
800	*P63/mmc*	a = b = 3.83473 c = 8.9117 V = 113.490 c/a = 2.32394	70.4 (2)	−3.5941 (7)
900	*P63/mmc*	a = b = 3.826(2) c = 8.899(7) V = 112.800 c/a = 2.3239	55.5 (6)	−0.0120 (9)
1000	*P63/mmc*	a = b = 3.83294 c = 8.90934 V = 113.355 c/a = 2.32441	93.4 (6)	7.8451 (5)
1100	*P63/mmc*	a = b = 3.83513 c = 8.91422 V = 113.547 c/a = 2.32436	80.00	1.2150 (6)
1200	*P63/mmc*	a = b = 3.83455 c = 8.91326 V = 113.500 c/a = 2.32446	89.2 (9)	11.2000 (0)

**Table 3 micromachines-17-00034-t003:** Scintillation yield (photons/MeV).

pH Value	Scintillation Yield (Photons/MeV) Relative to Gadox-Tb
**pH = 2**	6275
**pH = 5**	10,215
**pH = 8/Temperature °C**	
**600 °C**	779
**700 °C**	4997
**800 °C**	10,416
**900 °C**	9721
**1000 °C**	11,186
**1100 °C**	10,993
**1200 °C**	8845

**Table 4 micromachines-17-00034-t004:** Scintillation rise, decay times, average decay time, and estimated Coincidence Time Resolution (CTR) for different pH value.

Temperature (°C)	Rise Time (ns)	Fraction (%)	Short Decay (ns)	Fraction (%)	Late Decay (ns)	Fraction (%)	Average Decay (ns)	CTR (ps)
**2**	0.72	91.5	24.2	92	66.4	11	34.7	137
**5**	0.50	102	25.8	91	58.7	12	33.4	87
**8**	0.90	74	26	92	51.7	11	30.9	115

**Table 5 micromachines-17-00034-t005:** Scintillation rise, decay times, average decay time, and Coincidence Time Resolution (CTR) for different annealing temperatures.

Temperature (°C)	Rise Time (ns)	Fraction (%)	Short Decay (ns)	Fraction (%)	Late Decay (ns)	Fraction (%)	Average Decay (ns)	CTR (ps)
**600**	0.25	100	2.9	50	24.2	49	21.9	183
**700**	0.60	95	22.5	72	70.5	31	50.0	169
**800**	0.80	98	28.1	70	78.9	30	56.2	143
**900**	0.90	74	26	92	51.7	11	30.8	115
**1000**	0.71	101	27.0	94	68.5	9.	35.2	103
**1100**	0.64	104	28.7	96	65.6	7	33.8	96
**1200**	0.76	100	26.5	93	71.6	9	35.8	120

## Data Availability

No new data were created or analyzed in this study.

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
