# Peer review of "Micromachines2026, 17(1), 34;https://doi.org/10.3390/mi17010034"

_micromachines, 2025, doi:10.3390/mi17010034_

Round 1

Reviewer 1 Report

Comments and Suggestions for Authors

The article Photoluminescence and Scintillation Properties of Ce3+-Doped 2 GdBO3 Nanoscintillator Sensors: Effect of some synthesis parameters by Lakhdar Guerbous, Mourad Seraiche, Ahmed Rafik Touil, Zohra Akhrib and Rachid Mahiou deals with synthesis of nonsize crystalline cerium (Ce3+)-doped gadolinium orthoborate (GdBO3) phosphor for scintilate sensoring. 

The authors made a comprehensive investigation which could be interesting for the readers of  Micromashines.

Before a final decision is made about the publication, there are some questions that need to be clarified.

H3BO3 purity data is not provided. Corrections are required. Please specify the quality of the other reagents (acetylacetone, NH4OH, water?). You indicated high purity (99.9 wt%), which is required for scintillation materials. Therefore, it would be appropriate to provide an analysis of impurities in the final product.

There is a need for full description of all equipments , i.e (Manufacture, City, Country). Usually it is written as follows 

" The room temperature steady photoluminescence spectra were carried out using a LS-50B luminescence spectrometer (Perkin-Elmer Inc., MS, USA)"

The authors estimated the lattice parameters, average crystallite size, and macrostrain using the Williamson-Hall method (Table 1). What is the error in this method's determination of the coherence region, i.e., particle size? Currently, the Rietveld method is more reliable. A comparison of the data obtained by these two methods would be appreciated.

Figure 5 and 6. Transmittance >100% ??? Normalized or 3D image must be presented.

Table 2    9000 C must be corrected. Why at 900 C annealing the crystal size was the smallest? May be this is the reason for the highest PL intensity? Needs comments.

It is necessary to reconstruct Fig. 15. It is necessary to mark at least the axes.

The Abstract must be re-written with the emphasis on the main quantitative findings of the research

The same for the Conclusions.

Author Response

All comments and suggestions provided by the reviewer have been carefully addressed and incorporated into the revised manuscript. We also wish to thank the Editor and reviewers for their insightful remarks, which have significantly contributed to improving the quality of the manuscript.

We hope that our responses to the reviewers’ comments, along with the revisions and reorganization implemented in the manuscript, adequately address the concerns raised and provide a satisfactory and convincing clarification.

Reviewe #1

The article Photoluminescence and Scintillation Properties of Ce3+-Doped 2 GdBO3 Nanoscintillator Sensors: Effect of some synthesis parameters by Lakhdar Guerbous, Mourad Seraiche, Ahmed Rafik Touil, Zohra Akhrib and Rachid Mahiou deals with synthesis of nonsize crystalline cerium (Ce3+)-doped gadolinium orthoborate (GdBO3) phosphor for cintillate sensoring. 

The authors made a comprehensive investigation which could be interesting for the readers of  Micromashines.

Before a final decision is made about the publication, there are some questions that need to be clarified.

The authors sincerely thank the reviewer for the insightful comments and questions, which have enabled us to enhance the manuscript and deepen the discussion of the results obtained.

Comment 1. H3BO3 purity data is not provided. Corrections are required. Please specify the quality of the other reagents (acetylacetone, NH4OH, water?). You indicated high purity (99.9 wt%), which is required for scintillation materials. Therefore, it would be appropriate to provide an analysis of impurities in the final product.

Response 1:

Authors thanks the reviewer for the question ad comment:

The H3BO3 purity is 99.5 %, acetylacetone (99.00 %) ammonium hydroxide (NH4OH) (99.9%). High quality of de-ionized water (esistivity 18.2 MΩ·cm at 25 °C). 

All purities and qualities were cited in the revised manuscript

We agree with the reviewer concerning the analysis of impurities in the final product. In fact, this study aims to prepare GdBO₃: Ce³⁺ from lower-purity starting materials in order to achieve satisfactory results while reducing costs. We also assume that a high-temperature thermal treatment will eliminate nearly all organic residues.,

Comment 2. There is a need for full description of all equipments , i.e (Manufacture, City, Country). Usually it is written as follows " The room temperature steady photoluminescence spectra were carried out using a LS-50B luminescence spectrometer (Perkin-Elmer Inc., MS, USA)"

Response 2

All equipment were described as (Manufacture, City, Country) in the revised manuscript

Comment 3. The authors estimated the lattice parameters, average crystallite size, and macrostrain using the Williamson-Hall method (Table 1). What is the error in this method's determination of the coherence region, i.e., particle size? Currently, the Rietveld method is more reliable. A comparison of the data obtained by these two methods would be appreciated.

Response 3

We thank the reviewer for the question:

For further clarification this part has been added in the revised manuscript (page 4- Materials and Methods subsection):

The crystallite size and the lattice strain were evaluated using the Williamson–Hall (W–H) relationship [W–H].

  ………………………………………………(1)

Where, β represents the full width at half maximum, λ is the X-ray wavelength, θ denotes the diffraction angle, D is the effective crystallite size, and η indicates the effective strain.  In the Williamson–Hall method, it is assumed that the strain is isotropic and the crystallite size is uniform.  With plot of  against,   and a linear fit, the intercept provides the crystallite size, while the slope indicates the micro-strain. The standard error in the intercept of the W–H plot was used to estimate the uncertainty in crystallite size and micro-strain indicated in the table. “

Figure: Just An example of W-H linear adjustment performed in this work

In fact, the values reported in the original table were simply rounded. In the corrected table, the uncertainty value is noted in parentheses, according to the expression of uncertainty in the measurement.

The model adjustments were performed using HighScore Plus, the software associated with our equipment that employs the Rietveld method, which is recognized for its reliability. Notably, a comparison with FULLPROF software Suite produces results that are virtually identical.

As example: The Rietveld profile fits of X-Ray diffraction data of GdBO3, annealed at 900°C for 4 h

Comment 4. Figure 5 and 6. Transmittance >100% ??? Normalized or 3D image must be presented.

Response 4

Authors thanks the reviewer.  

The Figure 5 and 6. were normalized

Comment 5 . Table 2    9000 C must be corrected.

Response 5

thanks, the reviewer, corrected in the table

Comment 6. Why at 900 C annealing the crystal size was the smallest? May be this is the reason for the highest PL intensity? Needs comments.

Response 6

We thank the reviewer for the question and comment:

  • Why at 900 C annealing the crystal size was the smallest?

Indeed, this phenomenon was also observed at the beginning of our study.

Somewhat unexpectedly, the sample annealed at 900 °C exhibits a smaller crystallite size than those annealed at the other temperatures, with the exception of the 600 °C sample, despite all samples being prepared under identical conditions and annealed in the same furnace. Furthermore, XRD refinements were carried out to minimize possible errors; however, the same results were obtained.

In fact, we have attempted to explain this phenomenon; however, a convincing conclusion has not been reached, highlighting the need for further investigation in the near future.

For more clarification this part has been added in the page 8

“Unexpectedly, the sample annealed at 900°C exhibits a smaller crystallite size than those annealed at other temperatures, with the exception of the 600°C sample, despite all samples being prepared under identical conditions and annealed in the same furnace. It is well established that the growth of nanomaterials during heat treatment processes such as annealing, calcination, and sintering is governed by several interrelated thermodynamic and kinetic factors. These factors influence particle coalescence, grain growth, and phase transformations. Generally, higher temperatures favor larger crystallite sizes while reducing the density of structural defects. This unexpected behavior warrants further investigation.”

  • May be this is the reason for the highest PL intensity? Needs comments.

In fact, experimentally observed that sample annealed at 700 °C exhibits the most intense PL emission, which can be assigned to biggest crystallite size.   Nonetheless, the samples annealed at 800°C and 900°C show maximum PL Quantum Yield (PLQY) values the integration or bonding of the Ce3+ to the network

The following sentence is added to the revised manuscript (page 13):

“At temperatures above of 800°C, the PLQY decrement could be attributed to the integration or bonding of the Ce3+ to the network”

Comment 7. It is necessary to reconstruct Fig. 15. It is necessary to mark at least the axes.

Response 7   

We thank the reviewer

Axes were markets as recommended by the reviewer

Comment 8. The Abstract must be re-written with the emphasis on the main quantitative findings of the research

Response 8   

The abstract has been revised to provide a comprehensive overview of the study, highlighting the various aspects of the work and summarizing the key findings.

Comment 9. The same for the Conclusions.

Response 9

The conclusion has been rewritten to provide a comprehensive overview main quantitative results and findings of the research work

Reviewer 2 Report

Comments and Suggestions for Authors

Dears Lakhdar Guerbous, Mourad Seraiche, Ahmed Rafik Touil, Zohra Akhrib and Rachid Mahiou.

Concerning to your manuscript: micromachines-4032182

Photoluminescence and Scintillation Properties of Ce3+-Doped GdBO3 Nanoscintillator Sensors: Effect of some synthesis parameters

In this manuscript I found an investigation about the synthesis of gadolinium orthoborate phosphor (GdBO3) doped with Ce3+ by the sol-gel method and varying the pH and annealing temperatures in order to optimize the phosphor properties. In this way authors observe the effects of pH and annealing temperatures on the structural and spectroscopic properties. The vaterite phase obtained through the optimize methodology display emission bands through direct (photoluminescence) and X-ray (radioluminescence) excitation of the Ce3+ ions. The photoluminescence (PL) emission intensity increases with synthesis pH and decreases with annealing temperature. The scintillation yield under X-ray excitation, decay and timing kinetics parameters determined.

The manuscript was well writhe, organized and discussed, contains sufficient and pertinent references, the conclusion were correctly pointed and the synthesis methodology and the resulting material are interesting and applicable in different technological fields.  By this reasons I considered this research an interesting and acceptable document for micromachines/MDPI. Then, I only include minimal suggestions in the annex, in order to improve the comprehension of the manuscriprt.

  • In figure 1. The words Schematic diagram are redundant and could be writhe only> Scheme or sequence of synthesis.
  • In the pH effect section on page 5, authors say that. “in our case, the opposite behavior is observed: increasing the pH results in smaller crystallite sizes. This suggests that, rather than OH- ions, H3O+ ions may still play a dominant role in the reaction mechanism even at higher pH levels.” , however at higher pH, OH- can cause the rupture of some bonds and reduce the size of the polycondenzation products as occurs in an Oswald process
  • When authors writhe that: “the absence of water-related absorption bands in the FTIR spectra (sub-section 3.3) supports the hypothesis that water molecules are largely eliminated during the process, reinforcing the proposed mechanism.”

It is no clear to what process they refers, because the FTIR shows in Figure 5 refers to samples treated at 900oC.The sol-gel process, the hydrolysis-polycondenzation or what process? Logically, the annealing treatment causes the elimination of the physisorbed and chemisorbed water.

  • At temperatures above of 800oC, the QY decrement could be attributed to the integration or bonding of the Ce3+ to the network.
  • Authors abruptly finish their discussion without offer a deeper description of the structure of the systems analyzed, which could help to understand how the emission process occurs.
  • Furthermore, the presence of the acetylacetone could be responsible of some of the peculiarities observed in the results obtained, but authors only mentioned its function as chelating agent, but what was the chelated cation?

With the best regards,

The Reviewer

Author Response

All comments and suggestions provided by the reviewer have been carefully addressed and incorporated into the revised manuscript. We also wish to thank the Editor and reviewers for their insightful remarks, which have significantly contributed to improving the quality of the manuscript.

We hope that our responses to the reviewers’ comments, along with the revisions and reorganization implemented in the manuscript, adequately address the concerns raised and provide a satisfactory and convincing clarification.

Reviewe #2

Concerning to your manuscript: micromachines-4032182

Photoluminescence and Scintillation Properties of Ce3+-Doped GdBO3 Nanoscintillator Sensors: Effect of some synthesis parameters

 In this manuscript I found an investigation about the synthesis of gadolinium orthoborate phosphor (GdBO3) doped with Ce3+ by the sol-gel method and varying the pH and annealing temperatures in order to optimize the phosphor properties. In this way authors observe the effects of pH and annealing temperatures on the structural and spectroscopic properties. The vaterite phase obtained through the optimize methodology display emission bands through direct (photoluminescence) and X-ray (radioluminescence) excitation of the Ce3+ ions. The photoluminescence (PL) emission intensity increases with synthesis pH and decreases with annealing temperature. The scintillation yield under X-ray excitation, decay and timing kinetics parameters determined.

The manuscript was well writhe, organized and discussed, contains sufficient and pertinent references, the conclusion were correctly pointed and the synthesis methodology and the resulting material are interesting and applicable in different technological fields.  By this reasons I considered this research an interesting and acceptable document for micromachines/MDPI. Then, I only include minimal suggestions in the annex, in order to improve the comprehension of the manuscriprt.

The authors sincerely thank the reviewer for the insightful comments and questions, which have enabled us to enhance the manuscript and deepen the discussion of the results obtained.

Comment 1. In figure 1. The words Schematic diagram are redundant and could be writhe only> Scheme or sequence of synthesis.

Response 1:

As recommended, Replace by: 

“Figure 1. Sequence of steps for synthesizing GdBO₃:0.5mol% Ce³⁺ using the aqueous sol-gel method”

Comment 2. In the pH effect section on page 5, authors say that. “in our case, the opposite behavior is observed: increasing the pH results in smaller crystallite sizes. This suggests that, rather than OH- ions, H3O+ ions may still play a dominant role in the reaction mechanism even at higher pH levels.” , however at higher pH, OH- can cause the rupture of some bonds and reduce the size of the polycondenzation products as occurs in an Oswald process

Response 2:

We appreciate the reviewer comment regarding the potential role of OH⁻ ions in reducing the size of polycondensation products. Indeed, at higher pH, OH⁻ can promote bond cleavage. However, in our system, even at elevated pH, H₃O⁺-mediated hydrolysis and condensation pathways appear to dominate. The resulting kinetics favor a higher nucleation rate relative to crystal growth, leading to smaller crystallite sizes

The following part has been added in the revised manuscript (page 6, XRD analysis pH effect)

“More broadly, the observed decrease in crystallite size with increasing pH is likely associated with changes in precursor complexation, hydrolysis–condensation equilibria, and nucleation density inherent to rare-earth sol–gel systems, rather than with the dominance of a single ionic species.”

Comment 3. When authors writhe that: “the absence of water-related absorption bands in the FTIR spectra (sub-section 3.3) supports the hypothesis that water molecules are largely eliminated during the process, reinforcing the proposed mechanism.” It is no clear to what process they refers, because the FTIR shows in Figure 5 refers to samples treated at 900oC.The sol-gel process, the hydrolysis-polycondenzation or what process? Logically, the annealing treatment causes the elimination of the physisorbed and chemisorbed water.

Response 3:

-We thank the reviewer for the comment and suggestion,

-In fact, we just mean the heat treatment (annealing) process. It is corrected in the manuscript the sentence is replaced by:

“Furthermore, the absence of water-related absorption bands in the FTIR spectra (sub-section 3.3) confirms that water molecules are largely eliminated during the annealing process, reinforcing the proposed mechanism”

Comment 4 At temperatures above of 800oC, the QY decrement could be attributed to the integration or bonding of the Ce3+ to the network.

Response 4:

We thank the reviewer for the comment and suggestion,

As suggested the following sentence is added to the revised manuscript (page 13):

“At temperatures above of 800°C, the PLQY decrement could be attributed to the integration or bonding of the Ce3+ to the network”

Comment 5: Authors abruptly finish their discussion without offer a deeper description of the structure of the systems analyzed, which could help to understand how the emission process occurs.

Response 5:

This part has been added in the revised manuscript page (17) with additional diagram (Figure 15b) to explain more:

“Overall, these results indicate that the transfer efficiency improves as the annealing temperature increases. The schematic shown in the figure 15b illustrates the excitation energy transfer underlying the scintillation and photoluminescence mechanisms in GdBO₃:Ce³⁺.

Figure 15 (b): energy level diagram showing excitation energy transfer underlying the scintillation and photoluminescence mechanisms in GdBO₃:Ce³⁺.

Also, the authors thank the reviewer for pointing out the abrupt interruption in the discussion of CTR. This issue resulted from an inadvertent error in the version of the manuscript that was submitted.

This part has been added to complete the discussions

“A time-of-coincidence resolution (CTR) of less than 100 ps in time-of-flight positron emission tomography is expected to significantly enhance image quality [27]. It can be observed that the CTR values of all samples exceed 100 ps, with the exception of the sample annealed at 1100 °C. Moreover, a general improvement in CTR is observed with increasing annealing temperature, leading to enhanced temporal resolution.”

Added Reference : [27] D. Jeong, G. Chinn and C. S. Levin, "Time-Resolved X-Ray Luminescence Setup to Evaluate Thin Nanophotonic Super-Scintillator Samples," 2024 IEEE Nuclear Science Symposium (NSS), Medical Imaging Conference (MIC) and Room Temperature Semiconductor Detector Conference (RTSD), Tampa, FL, USA, 2024, pp. 1-1, doi: 10.1109/NSS/MIC/RTSD57108.2024.10654820.

Comment 6: Furthermore, the presence of the acetylacetone could be responsible of some of the peculiarities observed in the results obtained, but authors only mentioned its function as chelating agent, but what was the chelated cation?

Response 6:

Authors thanks the reviewer for the question:

In the sol-gel route, Acetylacetone is widely employed as a chelating agent, owing to its ability to form stable chelate complexes with rare-earth ions, such as gadolinium and cerium in the present system. It coordinates to these ions through its oxygen atoms, leading to the formation of chelated structures that may extend into chain-like arrangements. Moreover, acetylacetone preferentially coordinates with metal ions that are less stable, more reactive, or particularly susceptible to hydrolytic degradation, as chelation effectively stabilizes these ions by reducing their reactivity toward water.

For further clarification of the role of acetylacetone (AC), the following description has been added to the revised manuscript (page 2, Materials and Methods subsection):

“Acetylacetone (AC) is introduced to stabilize Gd³⁺ and Ce³⁺ cations through coordination via its oxygen atoms. This chelation moderates the hydrolysis and condensation kinetics, leading to the formation of a more homogeneous sol and gel network and promoting the uniform incorporation of the Ce³⁺ dopant. In addition, acetylacetone contributes to the control of crystallite growth and size. Consequently, by improving dopant distribution and microstructural homogeneity, acetylacetone indirectly enhances the luminescence efficiency and reproducibility of Ce³⁺-doped materials while reducing defect-related non-radiative quenching centers.”

Reviewer 3 Report

Comments and Suggestions for Authors

The paper reports the sol–gel synthesis of phase-pure hexagonal GdBO3 nanophosphors doped with 0.5 mol% Ce3+ and systematically examines how solution pH and annealing temperature influence their structural, photoluminescent, and scintillation properties, showing that higher pH improves crystallinity and enhances both PL and X-ray-excited luminescence, while annealing around 700–800 °C optimizes emission intensity, quantum yield, and light yield, with detailed decay-time analyses presented in Figures 11–15. However, the novelty of the work is limited: Ce-doped GdBO3 phosphors and sol–gel synthesis routes are well established, and the study mainly maps parameter-dependent trends rather than introducing new mechanisms or material concepts. Performance comparisons rely on internal sample-to-sample benchmarking without situating results against leading scintillators (e.g., GAGG:Ce, LYSO:Ce), making it challenging to effectively evaluate competitiveness. Reported quantum yields remain significantly below the literature benchmarks. Mechanistically, the paper attributes variations in PL/XEL to crystallite size, defect density, Ce3+/Ce4+ balance, and trap states; however, these explanations remain qualitative and lack direct evidence from structural probes (e.g., XPS for Ce oxidation states, TEM for defect density, and time-resolved PL to correlate traps). Overall, the paper needs rigorous revision before it can be considered for publication.

1. Although the authors present a systematic study on Ce3+-doped GdBO3 nanophosphors synthesized by an aqueous sol–gel method, several shortcomings limit the strength of the conclusions. Their central claim that pH and annealing temperature exert precise and predictable control over structural and optical properties is not quantitatively demonstrated. While XRD patterns at different pH values all show the vaterite phase, the authors’ mechanistic interpretation contradicts established sol–gel chemistry: they state that crystallite size decreases with increasing pH, yet assign this unexpected trend to a presumed dominance of H3O+ at all pH values without providing experimental validation. No independent measurements (e.g., zeta potential, precursor hydrolysis rate, gelation kinetics) are presented to support this ad hoc explanation, leaving the core structure–processing relationship insufficiently justified.

2. The discussion of annealing temperature effects suffers from similar weaknesses. The crystallite size and strain values fluctuate irregularly with temperature, rather than exhibiting systematic grain growth. Nevertheless, the authors attribute changes in optical output to improvements in crystallinity, without correlating PL intensity with quantitative structural descriptors. The XRD peak broadening is never modeled in detail beyond Williamson–Hall estimates, and no TEM images are provided to confirm nanoparticle morphology or validate size calculations. As a result, the causal link between annealing temperature, defect concentration, and optical behavior remains speculative rather than demonstrated.

3. In the FTIR analysis, the authors conclude that all samples exhibit identical vibrational signatures regardless of pH or annealing temperature. However, this uniformity undermines their earlier claim that pH strongly alters chemical structure during sol–gel processing. Suppose the coordination environment and borate-unit distribution are indeed unchanged. In that case, the paper does not explain how such an invariant local structure results in significant differences in PL intensity, XEL intensity, or scintillation kinetics. The lack of deconvolution, peak assignments with quantified shifts, or comparison to known borate structural models reduces the interpretive strength of the FTIR section.

4. The photoluminescence and radioluminescence data are also interpreted too loosely. The authors repeatedly attribute changes in emission intensity to crystallinity, unit-cell contraction, the Ce3+/Ce4+ ratio, and defect population; however, no direct measurements of valence state, defect density, or trap distributions are provided. For example, the proposed competition between Ce3+ and Ce4+ ions is mentioned, but no XPS or EPR spectra are shown to verify oxidation-state changes. Similarly, the explanation for intensity fluctuations based on cross-relaxation processes is unconvincing without Ce–Ce distance calculations or concentration-dependent studies. As a result, most mechanistic interpretations remain unsubstantiated hypotheses.

5. The scintillation analysis, although extensive, lacks proper benchmarking and validation. Light yield (LY) values are derived by comparison with Gadox-Tb, yet the authors do not report uncertainties, error propagation, or calibration details for the reference scintillator. The LY values vary widely with pH and annealing temperature without a clear structural rationale, and the claimed correlation between PL behavior and XEL behavior is contradicted by the data: PL intensity peaks at 700 °C, whereas XEL intensity maximizes at 800 °C, undermining the consistency of their interpretation. Moreover, the extracted rise and decay times show non-monotonic and inconsistent trends, but the authors treat these as systematic effects rather than acknowledging experimental limitations such as powder packing density, light-collection geometry, or sample heterogeneity.

6. Finally, several conceptual claims in the manuscript lack sufficient theoretical or experimental support. The authors suggest that their sol–gel route offers “precise control” over structure and performance, yet the observed property variations are irregular and not modeled quantitatively. The proposed relationship between the S/ηcoll ratio, traps density, and luminescence kinetics is described qualitatively but not validated by independent trap-probing techniques (thermoluminescence, defect spectroscopy). Throughout the paper, the mechanistic discussion often exceeds what the experimental data can justify, reducing confidence in the claimed processing–structure–property correlations.

Author Response

All comments and suggestions provided by the reviewer have been carefully addressed and incorporated into the revised manuscript. We also wish to thank the Editor and reviewers for their insightful remarks, which have significantly contributed to improving the quality of the manuscript.

We hope that our responses to the reviewers’ comments, along with the revisions and reorganization implemented in the manuscript, adequately address the concerns raised and provide a satisfactory and convincing clarification.

Reviewe #3

Comment 1. The paper reports the sol–gel synthesis of phase-pure hexagonal GdBO3 nanophosphors doped with 0.5 mol% Ce3+ and systematically examines how solution pH and annealing temperature influence their structural, photoluminescent, and scintillation properties, showing that higher pH improves crystallinity and enhances both PL and X-ray-excited luminescence, while annealing around 700–800 °C optimizes emission intensity, quantum yield, and light yield, with detailed decay-time analyses presented in Figures 11–15. However, the novelty of the work is limited: Ce-doped GdBO3 phosphors and sol–gel synthesis routes are well established, and the study mainly maps parameter-dependent trends rather than introducing new mechanisms or material concepts. Performance comparisons rely on internal sample-to-sample benchmarking without situating results against leading scintillators (e.g., GAGG:Ce, LYSO:Ce), making it challenging to effectively evaluate competitiveness. Reported quantum yields remain significantly below the literature benchmarks. Mechanistically, the paper attributes variations in PL/XEL to crystallite size, defect density, Ce3+/Ce4+ balance, and trap states; however, these explanations remain qualitative and lack direct evidence from structural probes (e.g., XPS for Ce oxidation states, TEM for defect density, and time-resolved PL to correlate traps). Overall, the paper needs rigorous revision before it can be considered for publication.

Response 1:

Authors thanks the reviewer for his comment, the following part has been added for more clarification in the introduction:

“Although numerous studies have reported on GdBO₃ doped with rare-earth ions such as Eu³⁺ and Pr³⁺, to the best of our knowledge, no investigations have addressed the synthesis of Ce³⁺-doped GdBO₃ via an aqueous sol–gel method. In this context, the present work represents the first systematic study of GdBO₃: Ce³⁺ prepared using the proposed sol–gel route. The study focuses on the synthesis of nanometric GdBO₃ :Ce³⁺ powders and provides an initial evaluation of their spectroscopic properties. Furthermore, the radioluminescence performance of the synthesized materials is benchmarked against that of the commercial Gadox:Tb³⁺ phosphor, highlighting their potential relevance for scintillation-related applications.”

Comment 2: Although the authors present a systematic study on Ce3+-doped GdBO3 nanophosphors synthesized by an aqueous sol–gel method, several shortcomings limit the strength of the conclusions. Their central claim that pH and annealing temperature exert precise and predictable control over structural and optical properties is not quantitatively demonstrated. While XRD patterns at different pH values all show the vaterite phase, the authors’ mechanistic interpretation contradicts established sol–gel chemistry: they state that crystallite size decreases with increasing pH, yet assign this unexpected trend to a presumed dominance of H3O+ at all pH values without providing experimental validation. No independent measurements (e.g., zeta potential, precursor hydrolysis rate, gelation kinetics) are presented to support this ad hoc explanation, leaving the core structure–processing relationship insufficiently justified.

Response 2:

We appreciate the reviewer’s pertinent and valuable comment. We recognize that the original version of the manuscript did not adequately elucidate the mechanisms by which pH and annealing temperature affect the structural and optical properties of Ce³⁺-doped GdBO₃ nanophosphors. Rather than claiming to provide a fully predictive quantitative model, our main objective was to highlight empirical trends regarding crystallite size, phase formation, and luminescence behavior as a function of synthesis parameters. We acknowledge that the manuscript's wording may have overestimated the level of "precise and predictable control.

We concur those additional analyses, including zeta potential measurements, hydrolysis kinetic studies, or gelation time assessments, would enhance the understanding of the underlying mechanisms. Although these measurements fall outside the scope of this study, they constitute significant areas for future investigation. Finally, despite these limitations, we emphasize that the structural trends observed from XRD analysis, together with consistent variations in photoluminescence and radioluminescence responses, provide a coherent and comprehensive understanding of how synthesis parameters influence the properties of GdBO₃:Ce³⁺ nanophosphors. We believe that the revised manuscript offers a more balanced and transparent discussion, thereby enhancing the validity of the conclusions.

The following part has been added in the revised manuscript (page 6, XRD analysis pH effect)

“More broadly, the observed decrease in crystallite size with increasing pH is likely associated with changes in precursor complexation, hydrolysis–condensation equilibria, and nucleation density inherent to rare-earth sol–gel systems, rather than with the dominance of a single ionic species.”

Comment 3:  The discussion of annealing temperature effects suffers from similar weaknesses. The crystallite size and strain values fluctuate irregularly with temperature, rather than exhibiting systematic grain growth. Nevertheless, the authors attribute changes in optical output to improvements in crystallinity, without correlating PL intensity with quantitative structural descriptors. The XRD peak broadening is never modeled in detail beyond Williamson–Hall estimates, and no TEM images are provided to confirm nanoparticle morphology or validate size calculations. As a result, the causal link between annealing temperature, defect concentration, and optical behavior remains speculative rather than demonstrated.

Response 3:

We acknowledge the reviewer’s observation that the crystallite size and microstrain values exhibit irregular variations with annealing temperature, rather than a systematic grain growth trend. We further recognize that attributing the observed changes in optical emission solely to enhanced crystallinity, in the absence of a direct correlation between photoluminescence intensity and quantitative structural parameters, may limit the robustness of our interpretation. In the present work, X-ray diffraction peak broadening was analyzed using Williamson–Hall approximations; however, we agree that more comprehensive line-profile modeling, together with complementary characterization techniques such as transmission electron microscopy (TEM) (not available at all in our country), would provide a more rigorous validation of nanoparticle morphology and crystallite size estimates. Consequently, the causal relationship between annealing temperature, defect concentration, and optical behavior cannot yet be established conclusively. These considerations identify important directions for future investigation, which we plan to address in subsequent studies.

Comment 4:  In the FTIR analysis, the authors conclude that all samples exhibit identical vibrational signatures regardless of pH or annealing temperature. However, this uniformity undermines their earlier claim that pH strongly alters chemical structure during sol–gel processing. Suppose the coordination environment and borate-unit distribution are indeed unchanged. In that case, the paper does not explain how such an invariant local structure results in significant differences in PL intensity, XEL intensity, or scintillation kinetics. The lack of deconvolution, peak assignments with quantified shifts, or comparison to known borate structural models reduces the interpretive strength of the FTIR section.

Response 4:

We appreciate the reviewer insightful comment concerning the FTIR analysis and its relevance to the proposed structure property relationships. We acknowledge that the overall similarity observed in the FTIR spectra across different pH values and annealing temperatures indicates that the primary vibrational modes and coordination environments of the borate network remain largely preserved within the detection limits of the technique. Accordingly, these FTIR results suggest that variations in pH during the sol-gel process do not induce significant alterations in the short-range connectivity of the borate framework.

We also acknowledge that this apparent structural consistency may appear to conflict with our earlier discussion regarding the influence of pH on material properties. However, we respectfully highlight that the observed variations in photoluminescence (PL), X-ray induced luminescence (XEL), and scintillation kinetics are likely attributable to more subtle structural and chemical factors such as changes in defect populations, hydroxyl group content, precursor complexation, and rare-earth ion distribution that do not produce significant shifts in the dominant FTIR bands. Since FTIR spectroscopy primarily reflects average local bonding environments, it may lack the sensitivity to detect these fine scale variations.

Furthermore, we concur that the interpretative robustness of the FTIR analysis would be enhanced by a more comprehensive examination, including peak deconvolution, quantitative evaluation of band positions and relative intensities, and direct comparison with established borate structural models. These aspects were not fully addressed in the current study but will be pursued in future work through integrated spectroscopic and structural methodologies. Accordingly, the manuscript has been revised to clearly define the scope and limitations of the FTIR analysis and to avoid overstating its implications regarding pH-induced structural modifications.

Comment 5:  The photoluminescence and radioluminescence data are also interpreted too loosely. The authors repeatedly attribute changes in emission intensity to crystallinity, unit-cell contraction, the Ce3+/Ce4+ ratio, and defect population; however, no direct measurements of valence state, defect density, or trap distributions are provided. For example, the proposed competition between Ce3+ and Ce4+ ions is mentioned, but no XPS or EPR spectra are shown to verify oxidation-state changes. Similarly, the explanation for intensity fluctuations based on cross-relaxation processes is unconvincing without Ce–Ce distance calculations or concentration-dependent studies. As a result, most mechanistic interpretations remain unsubstantiated hypotheses.

Response 5:

 We thank the reviewer for the thorough and constructive critique concerning the interpretation of the photoluminescence (PL) and radioluminescence (RL) data. We acknowledge that many of the mechanistic explanations proposed such as the influences of crystallinity, unit-cell contraction, the Ce³⁺/Ce⁴⁺ ratio, defect populations, and cross-relaxation processes are currently supported primarily by indirect evidence rather than direct experimental confirmation. Specifically, we agree that direct measurements of Ce oxidation states (e.g., via X-ray photoelectron spectroscopy or electron paramagnetic resonance), defect densities, trap distributions, and Ce–Ce interatomic distances were not included in the present study.

Future research will focus on directly investigating the Ce³⁺/Ce⁴⁺ ratio, defect states, and cross-relaxation mechanisms using techniques such as X-ray photoelectron spectroscopy (XPS), electron paramagnetic resonance (EPR), and concentration-dependent photoluminescence studies. These approaches are expected to provide more quantitative mechanistic insights and thereby strengthen the established causal relationships among structural parameters, defect chemistry, and luminescence behavior.

Comment 6:  The scintillation analysis, although extensive, lacks proper benchmarking and validation. Light yield (LY) values are derived by comparison with Gadox-Tb, yet the authors do not report uncertainties, error propagation, or calibration details for the reference scintillator. The LY values vary widely with pH and annealing temperature without a clear structural rationale, and the claimed correlation between PL behavior and XEL behavior is contradicted by the data: PL intensity peaks at 700 °C, whereas XEL intensity maximizes at 800 °C, undermining the consistency of their interpretation. Moreover, the extracted rise and decay times show non-monotonic and inconsistent trends, but the authors treat these as systematic effects rather than acknowledging experimental limitations such as powder packing density, light-collection geometry, or sample heterogeneity.

Response 6:

We thank the reviewer for the thorough assessment of our scintillation analysis. We acknowledge that the reported light yield (LY) values, obtained by comparison with a Gadox-Tb reference, lack explicit reporting of uncertainties, error propagation, and calibration details for the standard. In future studies, we plan to include full uncertainty analyses and more detailed calibration procedures to enhance the rigor and reproducibility of LY measurements.

We also recognize that the observed variations in LY and the differences between photoluminescence (PL) and X-ray excited luminescence (XEL) intensities with PL peaking at 700 °C and XEL at 800 °C indicate that the correlation between PL and XEL behavior is not straightforward.  

For more clarification this part has been added in the revied manuscript (page.15):

“The observed variations in light yield (LY), together with the differences in photoluminescence (PL) and X-ray excited luminescence (XEL) intensities, where PL exhibits a maximum at 700 °C and XEL at 800 °C, suggest that the relationship between PL and XEL behavior is not straightforward. These discrepancies may originate from distinct excitation mechanisms, differences in penetration depth, or non-radiative processes that are not fully reflected in PL measurements alone.”

With respect to the rise and fall times, we concur that the observed non-monotonic trends likely arise from a combination of the intrinsic scintillation dynamics and experimental factors, including powder packing density, sample heterogeneity, and light collection geometry.

We emphasize that all measurements of PL and XEL were conducted under strictly identical experimental conditions to ensure a robust and reliable comparison.

Comment 7:  Finally, several conceptual claims in the manuscript lack sufficient theoretical or experimental support. The authors suggest that their sol–gel route offers “precise control” over structure and performance, yet the observed property variations are irregular and not modeled quantitatively. The proposed relationship between the S/ηcoll ratio, traps density, and luminescence kinetics is described qualitatively but not validated by independent trap-probing techniques (thermoluminescence, defect spectroscopy). Throughout the paper, the mechanistic discussion often exceeds what the experimental data can justify, reducing confidence in the claimed processing–structure–property correlations.

Response 7:

 We thank the reviewer for highlighting these important points regarding the conceptual claims in our manuscript. We acknowledge that statements suggesting “precise control” over structure and performance via our sol gel synthesis should be interpreted within the context of the observed variations, which indeed exhibit some irregularity and are not captured by quantitative modeling.

With respect to the proposed relationships between the the S/ηcoll ratio, trap density, and luminescence kinetics, we acknowledge that the present analysis remains qualitative and has not been independently verified through direct trap-probing techniques, such as thermoluminescence or defect spectroscopy.

In future work, we will explicitly present these mechanistic interpretations as hypothesis-driven insights rather than definitive conclusions, thereby clarifying their exploratory nature while guiding further experimental validation.

Round 2

Reviewer 1 Report

Comments and Suggestions for Authors

Thank you for the clear answers.

The authors answered all the questions and made full corrections of tte mistakes.